# New Thermo-Reflective Coatings for Applications as a Layer of Heat Insulating Materials

**DOI:** 10.3390/ma15165642

**Published:** 2022-08-17

**Authors:** Elżbieta Malewska, Aleksander Prociak, Laima Vevere, Edgars Vanags, Marcin Zemła, Katarzyna Uram, Mikelis Kirpluks, Ugis Cabulis, Mirosław Bryk

**Affiliations:** 1Department of Chemistry and Technology of Polymers, Cracow University of Technology, Warszawska 24, 31-155 Cracow, Poland; 2Latvian State Institute of Wood Chemistry, Dzerbenes 27, LV-1006 Riga, Latvia; 3Implementation Company “Damiton”, Mirosław Bryk, Stadnicka 1c, 32-410 Dobczyce, Poland

**Keywords:** reflective materials, coatings, thermal properties

## Abstract

This paper presents new thermo-reflective coatings with different properties. Basic, anti-corrosion and self-extinguishing coatings were analyzed. The coatings were obtained with a thickness varying from 1 to 3 mm. The coatings were subjected to detailed tests assessing their physical-mechanical properties, i.e., tensile strength, abrasion, pull-off test, water absorption, vapor permeability and thermal properties, i.e., the thermal performance of the reflective coatings, thermal transmittance, thermogravimetric analysis, differential scanning calorimetry, as well as thermomechanical analysis and thermal conductivity. In addition, the possibility of using such coatings in a wide range of temperatures and during application to various materials used as a substrate, i.e., concrete, metal and rigid polyurethane foam, was tested. The thermal analysis of coatings revealed that materials are stable to temperatures above 200 °C, there are no thermal transitions in the negative temperature region and shrinking in low temperatures is minimal (less than 0.5%). From the data obtained within the framework of this study, it can be concluded that anticorrosive, basic and self-extinguishing coatings are eligible for thermo-insulation applications in temperatures up to 200 °C.

## 1. Introduction

The interest in thermo-reflective coatings began in the 1970s. Since then, many researchers have been studying their basic properties and their possible applications. A thermo-reflective coating is a material that is applied to other surfaces to achieve favorable properties when exposed to solar radiation. Thermo-reflective materials, also called cool coatings, are characterized by a high reflection of solar radiation. They are highly reflective, absorbs less heat. The surface with such a coating has a lower temperature compared to surfaces that do not have thermo-reflective materials [1]. In addition to individual benefits for home or road users, the use of thermo-reflective coatings also provides benefits on a global scale. For many years, an effect of urban agglomerations heating to temperatures higher than the surrounding areas has been observed, which is known as urban heat island effect (UHI). UHI is caused by a large concentration of artificially constructed structures that absorb more solar radiation than natural surfaces. The use of solar reflecting coatings can allow the reduction of the temperature of urban surfaces, i.e., roofs, roads and pavements, contributing to the reduction of UHI, and thus to energy savings, which in large agglomerations is used to power air conditioners, and can increase the comfort of its inhabitants [2,3,4].

Solar radiation that falls on surfaces not protected by thermo-reflective coatings is absorbed by them. The absorption of solar radiation causes the surface temperature of a building to rise above the ambient temperature. If this heat cannot be reflected outside, it radiates through the roof, ceiling and thermal insulation layers into the building interior. This situation causes a heat load on the building, especially in large industrial buildings and supermarkets [5,6].

The use of coatings with high reflectance of visible light and near infrared is currently the most effective way to reduce the UHI phenomenon. The literature describes many possible surfaces on which thermo-reflective coatings can be applied, such as bricks, ceramic and concrete roof tiles, asphalt, etc. [7,8,9,10,11,12]. The roofs and facades of buildings are covered with thermo-reflective coatings. They are most often used in the form of paints and waterproof membranes. The use of natural materials, such as marble and gravel in light colors, also gives good results [1]. Joudi et al. [13] report that in milder climates the best results are obtained by using a reflective coating both outdoors and indoors, while in cooler climates only indoors.

Apart from highly reflective and emissive light colored materials, cool colored materials are also described in the literature, i.e., colored materials with increased near-infrared reflectance, dynamic materials with properties that change their reflectivity depending on temperature and solar radiation, and materials that undergo phase change [12].

The influence of the coating color on the surface temperature was presented by Hernández-Pérez [14] who used three types of reflective coatings: red roof, gray roof and white roof. Due to the ability to reflect solar radiation, the temperature of white coatings was 10 °C lower than that of the gray coatings. 

In order to obtain thermo-reflective coatings, mixtures containing several components are usually used. The main components of reflective paints are functional materials, resins combining ingredients and special additives [15]. Reflective, insulating and radiation materials can be used as functional materials. Often, hollow microspheres made of ceramics, glass or polymers are added for this purpose [16,17]. Resin in the paint must have high transparency [18].

In the literature, the authors focus primarily on determining the ability to absorb radiation by coatings but do not describe the physico-mechanical properties, thermal conductivity and resistance to weather conditions [2,19,20,21,22,23]. Many researchers focus on the analysis of temperature changes inside buildings and on the surface that is exposed to sunlight. Kusis et al. described the possibility of using a new transparent binder that was combined with the aggregate. The described connection minimizes the local overheating of the surface. Thanks to the application of this composition on the tested surface, the temperature was reduced by 9.2 °C [24]. A similar effect was obtained by Zhang et al. using a coating composed of a zirconia (ZrO_2_)-embedded polydimethylsiloxane, lowering the surface temperature by 10.9 °C [25]. However, these all analyzes do not verify the mechanical properties of coatings, their durability and adhesion to various materials used in the construction of buildings and structures [18]. In the professional literature, the authors mainly focus on the analysis of the thermo-reflectivity of coatings, ignoring other features of materials necessary in construction. In addition, there is no information about the possibility of using coatings for industrial installations operating at low temperatures, such as pipelines and tanks. Therefore, the obtained reflective coatings were subjected to a series of analyzes of their physical and mechanical properties. The possibility of applying coatings to materials, such as concrete, metal and polyurethane foam, was tested. The novelty of the conducted research is the combination of favorable properties of reflective coatings with anti-corrosion and self-extinguishing properties. Additionally, the authors analyzed the possibility of using the tested coatings at temperatures below zero.

## 2. Materials and Methods

### 2.1. Characteristic of Raw Materials

Three types of coating compositions were provided by the Damiton company and used in this research: the base composition (RC-B), the anti-corrosion composition (RC-A) and the self-extinguishing composition (RC-S). The basic characteristics of the pastes are given in Table 1. These detailed compositions are the expertise of the Damiton company. The main ingredient of the pastes is titanium dioxide (Tiona 595 produced by Cristal Pigment UK Ltd., Grimsby, UK) and microspheres up to 100 microns diameter supplied by 3M Company (St. Paul, MN, USA). Additionally, the provided pastes are a product based on polyvinyl–acrylic dispersion with the addition of dispersants, emulsifiers, pigments, fillers, pH stabilizers, thickeners, preservatives and rheology modifiers. In addition, the anti-corrosion paste contains anti-corrosive agents (Ascotran H10 produced by ASCOTEC^®^, Düsseldorf, Germany, which works in the wet phase after application and Asconium 142DA—works in a dried coating). The self-extinguishing paste contains a flame retardant, namely aluminum hydroxide (ATH), supplied by Brenntag Polska Sp. z o.o. (47-224 Kędzierzyn-Kozle, Poland).

### 2.2. Coatings Preparation Method

The coatings were prepared from the blend by applying them to the surface to obtain layers with a thickness of 1.0, 1.5, 2.0, 2.5 and 3.0 mm, respectively. Additionally, the blend was diluted with 2, 4 and 6% water, and coatings with thickness of 1.0 and 3.0 were prepared. The blend was applied to the surface by hand with a finishing spatula. The coatings were allowed to dry for 3 days under room conditions, that is, prevailing conditions in an air-conditioned laboratory, i.e., 21 °C, about 1000 hPa and at a humidity of about 60%. After this time, the surface of the coating was leveled by grinding off irregularities.

### 2.3. Measurement Methods

#### 2.3.1. Methods of Measuring Thermal Properties of Coatings

In order to investigate the thermal performance of the reflective coatings, thermal transmittance, thermogravimetric analysis, differential scanning calorimetry, as well as thermomechanical analysis and thermal conductivity were applied.

The thermal conductivity was found using a Fox200 Lasercomp heat flow instrument (New Castle, DE, USA). The temperature of the cold plate was 0 °C, while the temperature of the warm plate was chosen to be 20 °C in order to achieve an average temperature of 10 °C. The thermal conductivity was determined 7 days after the preparation of a coating. 

For the thermogravimetric analysis of thermo-reflective coating samples, a Mettler Toledo TGA/SDTA 851 thermogravimeter was used (Columbus, OH, USA). Three thermo-reflective coating samples were tested: RC-B, RC-A and RC-S. Each sample was dried and ground to a powder. The weight of the tested samples was about 8 mg. The samples were heated to 800 °C (10 °C/min) in both air and nitrogen media. 

For the differential scanning calorimetry analysis of thermo-reflective coating samples, Mettler Toledo DSC 823 was used. Each sample was dried and ground to a powder. The weight of the tested samples was about 5 mg, and for testing, aluminum crucibles with pins were used. The samples were cooled from 25 °C to −150 °C (10 °C /min) and heated from −150 °C to 200 °C (10 °C/min).

#### 2.3.2. Methods of Measuring of Thermal Transmittance

For the thermal transmittance experiment, two aluminum plates of exact size and thickness (500 mm × 500 mm × 2.05 mm) were used. The first aluminum plate was used as a reference. On one side, it was covered by a paper tape to prevent the high infrared irradiation reflectance of the bare aluminum surface. The second aluminum plate was covered with a layer of about 4.50 mm of thick thermo-reflective basic coating (Figure 1). Both plates were tested under the following conditions.

The aluminum plate was mounted on two stands, ensuring a 90° angle against the floor. As a heat source heat gun (STEINEL HG 2320 E) with a 9 mm inner diameter tip was used. The airflow settings of the heat gun were preset to 250 °C and 50% airflow intensity. The heat gun was mounted on a stand, ensuring a 20 mm distance between the tip of the heat gun and the aluminum plate’s bare side (Figure 2). To detect thermal transmittance, a thermal imager (testo 880, infrared lens 32°10 mm f/1.0, Sparta, NJ, USA) was mounted on a tripod directed straight at the opposite (covered) side of the aluminum plate, ensuring 70 cm distance between the plate and the lens. The test was performed by switching on the heat gun and taking thermal images every 30 s for 5 min. The thermal images afterwards were processed using IRSoft 4.6 software to determine the hottest spot and average temperatures of both plates during the heating.

#### 2.3.3. Methods of Measuring Physical and Mechanical Properties of Coatings

For the thermo-mechanical analysis of thermo-reflective coating samples, a Linseis TMA PT was used. Each sample was cut in a rectangular cuboid with h~2 cm. The sample was cooled from 20 °C to −160 °C (3 °C/min) and then heated to 50 °C (3 °C/min). 

In order to determine the mechanical properties of the coatings, a tensile strength, abrasive wear and a pull-off test were performed. The tensile strength was found to be in line with PN-EN ISO 527. The tensile strength test was carried out at a speed of 2 mm/min. The measuring section was 30 mm long. The maximum stress, stress at break, elongation at maximum stress, elongation at break and Young’s modulus were determined. Measurements were carried out on 5 samples. 

The abrasive wear of the coatings was measured with a Schopper apparatus in line with PN-ISO 4649. For the test, 240 grit sandpaper was used. Each sample was tested for 8 s. The material volume change (Δ*V*) was calculated using the formula:$$\Delta V=\frac{m_{1}-m_{2}}{d}$$

*m*_1_—sample weight before testing, g

*m*_2_—sample weight after testing, g

*d*—sample density, g/mm^3^

Pull-off tests were conducted with Pull-Off type TPO-W10 machine produced by the Multiserw Morek company (Marcyporęba, Poland). The paste was applied to three types of surfaces: concrete, a metal plate made of S235 black steel and a rigid polyurethane foam with an apparent density of 40 kg/m^3^. The coatings were glued to the round tear-off plate with a two-component adhesive based on epoxy resin (trade name: POXIPOL). The tear-off plates were round, made of stainless steel with a diameter of 50 mm.

The dimensional stability was calculated using the formula recommended in the PN-92/C-89083 standard. The dimensional stability of the coating was tested 24 h after sample conditioning at +70 °C, 95% humidity and −25 °C.

The analysis of the aging process was carried out for a month at a temperature of 70 °C, 50% humidity and under lamps emitting visible light with a power of 24 W and a color of 4000 K. The samples were observed for color change and changes in mechanical properties after the aging process.

Water absorption was calculated using the formula recommended in the PN-EN 12087 standard. The determination of water vapor transmission properties, using the cup method, was carried out in accordance ISO 7783:2018.

The flammability of selected coatings was analyzed by determining the oxygen index (OI) value in accordance with PN-EN ISO 4589-2:2006.

#### 2.3.4. Analysis of the Chemical Structure of the Coating

The chemical structures of the three different thermo-reflective coatings were analyzed by FTIR (Fourier transform infrared spectroscopy) data obtained with a Thermo Fisher Nicolet iS50 spectrometer at a resolution of 4 cm^−1^, 32 scans (Waltham, MA, USA). The FTIR data were collected using an attenuated total reflectance technique with ZnSe and Diamond crystals. The samples of thermo-reflective coatings were pressed down on ATR diamond prism, and the spectra were collected.

## 3. Results and Discussion

### 3.1. Chemical Structure Analysis Results

In the conducted research, three types of coatings were prepared: base (RC-B), self-extinguishing (RC-S) and anti-corrosive (RC-A). FTIR analysis was performed for the analyzed samples.

The FTIR spectra of the samples RC-B, RC-S and RC-A are depicted in Figure 3. The absorption maximum at 1731 cm^−1^ represents ester C=O stretching, which refers to acetate ester groups in poly(vinyl acetate) polymer. The highly intensive absorption maximum at 1017 cm^−1^ is characteristic for the Si-O stretching vibration, referring to silicon dioxide, which is the main component of coatings. The described peaks are the most intense and come from the basic components found in all the pastes described, therefore there are no significant differences between the analyzed coatings.

### 3.2. Thermal Properties Analysis Result

The basic feature of materials used in construction for thermal insulation is their thermal conductivity. It was decided to examine whether the analyzed reflective materials are also characterized by thermal insulation properties. Currently, the best thermal insulation materials are closed-cell polyurethane foams, which can have a thermal conductivity coefficient of 22 mW/m·K [26,27]. On the other hand, currently the most commonly used thermal insulation in construction is still an expanded polystyrene board, the thermal conductivity of which is about 35 mW/m·K [27]. 

The measurements of the thermal conductivity were carried out 7 days after the preparation of the thermo-reflective coating samples. The previous analysis showed that thermal conductivity of samples stabilizes after 3 days of drying. In addition, it was observed that the thickness of the prepared samples significantly influences the value of the thermal conductivity. If the thickness of the sample is thinner, the thermal conductivity is lower. The figure below (Figure 4) shows a SEM photo of an exemplary cross-section of thermo-reflective coating. The photo shows that the structure of the coating is homogeneous throughout the cross-section.

The most favorable values were obtained for a thickness of 1 mm, for which the thermal conductivity was approx. 34 mW/m·K for the RC-B coatings. While for the 3 mm thick sample, the thermal conductivity was approximately 50 mW/m·K. The self-extinguishing coatings had the highest thermal conductivity among the analyzed coatings. For the thickest 3 mm sample, the value of the thermal conductivity was approximately 70 mW/m·K, which was about 30% higher than in the case of RC-B and RC-A samples. For the thinnest samples, i.e., 1 and 1.5 mm, the values of the thermal conductivity are the same for all types of coatings at about 35 mW/m·K (Figure 5). The obtained values of the thermal conductivity coefficient for the analyzed coatings are promising compared to thermal insulation plasters used to face the problems of energy efficiency in construction available on the market, the value of which ranges from 55 to even 110 mW/m·K [28,29]. Bao et al. prepared polyacrylate/hollow spherical TiO_2_ composite film, the thermal conductivity of which even reached 200 mW/m·K [16].

In order to prepare coating samples of a low thickness of 1 and 3 mm without defects, an additional amount of water was used to reduce the viscosity of the thermo-reflective paste. The ability to adequately reduce the viscosity of the composition by adding water is advantageous in terms of airless application.

The thermal conductivity (after 7 days) of materials obtained with the pastes diluted by 2, 4 and 6% of water is shown in Figure 6. It was observed that the addition of water did not change the value of the thermal conductivity. Only in the case of the RC-S coating sample with a thickness of 3 mm, was a favorable decrease in the value of the thermal conductivity observed.

The samples of coatings were thermo-gravimetrically analyzed in both nitrogen and air media using the dynamic TGA method to determine the thermal stability properties of the thermo-reflective coatings. The derivatization of the obtained TGA curves allowed the determination of the thermal degradation steps of the material as well as the temperature at which the degradation occurs most rapidly. The results of thermal stability interpretation by TGA are summarized in Table 2. T_OS_ represents initial offset temperature, while T_5_ to T_20_ represent the temperatures of 5 wt.% mass loss, etc.

It is noticeable that the thermal degradation and thermo-oxidative degradation of all thermo-reflective coatings occurs in two degradation steps. The first step of the thermal degradation of thermo-reflective coatings in inert media (Figure 7), according to previous poly(vinyl acetate) (PVA) thermal degradation studies, is the deacetylation of PVA, which is transformed into unsaturated residue—polyene. Regarding RC-S coating, the first step of thermal degradation occurs at a lower and broader temperature range (215–320 °C). The first degradation step for RC-S coating is significantly more intense than it is observable for RC-B and RC-A coatings in temperature ranges of 255–310 °C and 250–298 °C, respectively. The more pronounced mass loss in the first step of thermal degradation for RC-S coating could be explained by the evaporation of added flame retardants.

The second thermal degradation step of thermo-reflective coatings in nitrogen media can be explained by polyene thermal degradation, subsequently leaving around 70% of silicon dioxide as an incombustible residue, which is the main component of the coatings. The second thermal degradation step for RC-S, RC-A and RC-B coatings occurs in temperature ranges of 320–450 °C, 320–490 °C and 300–475 °C, respectively.

The comparison of the obtained results in an inert and air media (Figure 8) reveals that the first step of thermal degradation is more expressed in air media, indicating oxidative degradation beside deacetylation. Additionally, it can be seen that the first and the second degradation steps in the air are converging and less distinctive for all coatings. Both oxidative degradation steps for RC-S coating occurs within a temperature range of 215–430 °C, for RC-A coating within 220–430 °C and for RC-B within 225–435 °C. It is noticeable that the presence of air decreases the onset temperature RC-A and RC-B coatings by 20–30 °C. For RC-S coating, the onset temperature remains unchanged.

Overall, from the TGA data obtained, it can be concluded that the thermo-reflective coatings are not recommended for use in temperatures higher than 200 °C.

DSC analysis is a widely used technique for the structural investigation of different polymeric materials. The sample RC-B of thermo-reflecting coating had the lowest glass transition (Tg) temperature (Table 3), followed by the RC-A sample of thermo-reflecting coating. No thermal transitions were detected for the sample S of thermo-reflecting coating. This may indicate some molecular bond formation between the B coating formulation and the additives that provide the anti-corrosiveness and self-extinguishing. There were no thermal transitions in the negative temperature region (Figure 9), indicating that materials are suitable for use in cryogenic temperatures. 

The principle of the dilatometry method is to measure the materials volume variation as a function of the temperature change. Dilatometry was applied to evaluate thermo-reflecting coatings properties, such as relative expansion and coefficient of thermal expansion (CTE) (Table 4). As the temperature decreased, all three samples shrank. The RC-S had slightly lower relative expansion and CTE than other coatings (Figure 10). It could be related to changes in the coating’s formulation. Thermo-reflecting coatings’ relative expansion and CTE were lower than for polyurethane foam (Figure 11). This can help reduce multi-layer cryogenic isolation expansion/contraction as the temperature changes.

### 3.3. Thermal Transmittance Analysis Results

A test quite often used in the analysis of reflective coatings is the measurement of the temperature of the irradiated surface. The purpose of using thermo-reflective coatings is to reflect the radiation and thus lower the temperature on the building surface and inside [29,30]. From the acquired thermal image data, it can be seen that the thermo-reflective coatings have a considerable influence on the heat transport. The hottest spot on the uncoated aluminum plate after 5 min of heating reached 86.3 °C, while the plate with the 4.5 mm thick thermo-reflective B coating reached only 65.3 °C, thus proving the thermal insulating properties of the thermo-reflective coating (see Figure 12). 

During the measurement, photos were also taken with a thermal imaging camera. The photos below clearly illustrate the temperature changes taking place on the surface of the aluminum plate. The thermal images of the uncoated and coated plates during the heating are compiled in Figure 13.

### 3.4. Restults of Pysical and Mechanical Propeties Analysis

A number of tests of mechanical properties were performed for the analyzed coatings. Tensile strength, Young’s modulus, abrasive wear, pull-off tests and dimensional stability were examined. The determination of mechanical properties is necessary to determine their suitability for specific applications. If the roof is covered with a coating, it should be checked whether the coating can be walked on, whether it will delaminate from the substrate and how weather conditions will affect it.

The RC-S coating had the highest tensile strength. The RC-A coating with water addition was characterized by the lowest maximum stress and the lowest stress at rupture. Materials with 6% water addition had lower strength than coatings without water addition. The introduction of water into the thermo-reflective compositions resulted in a reduction in elongation at maximum stress and a reduction in the elongation at break of coatings compared to non-water based materials (Figure 14). The value of Young’s modulus for the coatings was similar, except for the S coating with the addition of water, the modulus of which was much higher (Figure 15).

The accelerated aging process of the coatings was carried out in a climatic chamber ensuring constant temperature, humidity and light conditions. After one month of aging, no optical changes were observed on the surface of the coatings. The material did not turn yellow or become cracked (Figure 16). The samples were also analyzed for mechanical properties. The results are shown in Figure 11 and Figure 12. The aging process resulted in an increase in the tensile strength of the material by up to 100%, while at the same time reducing the value of elongation at break. As the coating ages, the residual water, which exhibits a plasticizing effect, evaporates, causing the material to harden.

The RC-B coatings were characterized by the lowest abrasion compared to other compositions. The abrasive consumptions of the tested coatings applied with 6% water were similar to the coatings obtained from composition without dilution. The self-extinguishing coatings had the lowest abrasion resistance (Figure 17).

The following types of cracks may occur when the coatings are pulled off, which are considered a normal test result: adhesive crack—a crack at the interface between the coating and the surface of substrates. The adhesion is then equal to the test result; cohesive crack 1—crack in the coating itself—then the adhesion is greater than the test result; cohesive crack 2—crack in the substrate material—the adhesion is greater than the test result. The average adhesive strength values are shown in Figure 18. 

During the test, it was observed that the breaking of the coatings on the concrete surface always took place inside the coating (cohesive crack 1). In the case of using a metal plate, the peeling of the RC-B coating was observed at the joint with the metal plate (adhesive crack), while RC-A and RC-S coatings were torn inside the coating (cohesion crack 1). The peeling of the RC-B coating from the polyurethane foam took place at the contact of the surface with the coating (adhesive cracks), while the RC-A and S coatings caused cracks in the material substrate. The Figure 19 shows the appearance of the surface after the pull-off test.

Additionally, the oxygen index analysis was performed on the samples in order to check the fire resistance of the obtained coatings. All samples are self-extinguishing under atmospheric conditions, which positively affects their fire safety (Table 5). The RC-S coatings are characterized by the highest value of the oxygen index, which is over 40%. On the other hand, coating RC-B and RC-A had an oxygen index of about 23%. Table 4 additionally summarizes the values of apparent density, water absorption and water vapor diffusion resistance factor of the tested coatings. The values of the apparent density of tested coatings are correlated to the density of used pastes and content of light fillers (Table 1), while differences in water vapor diffusion resistance factor are additionally caused by the flame retardants used in the S coating. In both applied low and high temperatures, all tested coatings are dimensionally stable.

## 4. Conclusions

The conducted analyzes of thermo-reflective coatings show that the modification of the base coating to obtain the desired properties, such as self-extinguishing or anti-corrosive properties, did not change the most important functional properties of the coatings. The addition of ATH flame retardant allows an oxygen index above 40% to be obtained. The dimensional stability remains unchanged for all samples and is below 1%, as is the water absorption of approximately 22%. In addition, the modified coatings were characterized by a greater strength of adhesion to all analyzed substrates, with the best adhesion to concrete and the weakest to polyurethane foam, which, however, was related to the lowest strength of the polyurethane substrate that was torn. In the case of modified materials, the abrasion wear and water vapor diffusion resistance have also improved, which guarantees better moisture exchange with the environment compared to the base coat.

To evaluate the eligibility of thermo-reflective coatings for thermal insulation purposes, various thermal analysis methods were applied. The thermal transmittance experiment of basic coating showed decreased heat transition which approves thermo-insulating properties of the coating. The use of this type of thermo-reflective coatings allows the surface temperature to be reduced by approximately 20 °C.

In addition, it was found that the obtained coatings are characterized by a favorable thermal conductivity, which, for coatings with a thickness of 1 mm, was below 40 mW/m·K.

The thermal analysis of coatings revealed that materials are stable up to temperatures of 200 °C, there are no thermal transitions in the negative temperature region and shrinking in low temperatures is minimal (less than 0.5%).

From the data obtained within the framework of this study, it is concludable that anticorrosive, basic and self-extinguishing coatings are eligible for thermo-insulation applications in temperatures up to 200 °C.

The coatings had appropriate mechanical properties and resistance to weather conditions. The use of such coatings as a part of sandwich panels can additionally increase fire safety.

## Figures and Tables

**Figure 1 materials-15-05642-f001:**
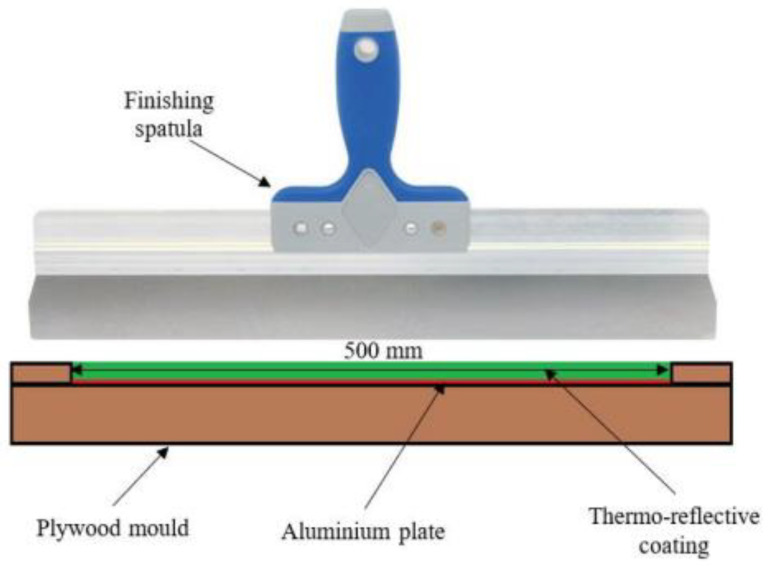
Application of thermo-reflective coating.

**Figure 2 materials-15-05642-f002:**
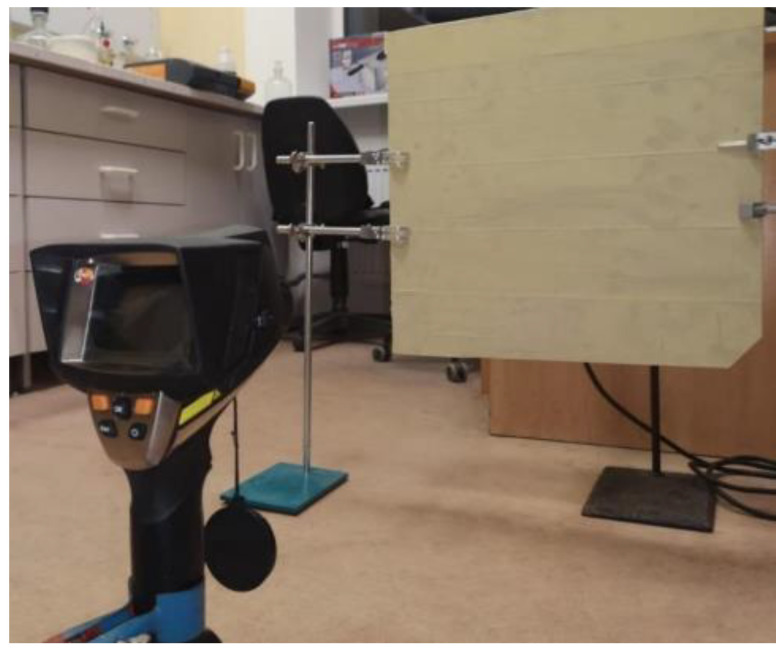
The setup of thermal transmittance experiment.

**Figure 3 materials-15-05642-f003:**
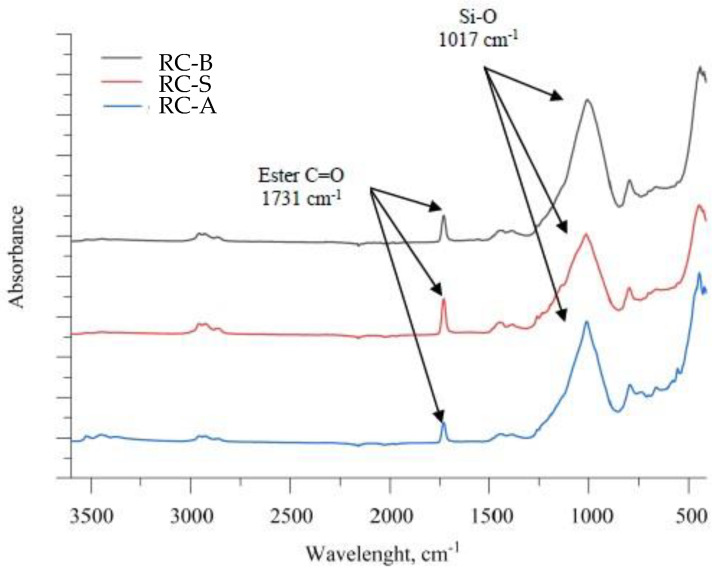
FTIR spectra of the samples RC-B, RC-A and RC-S of thermo-reflective coatings.

**Figure 4 materials-15-05642-f004:**
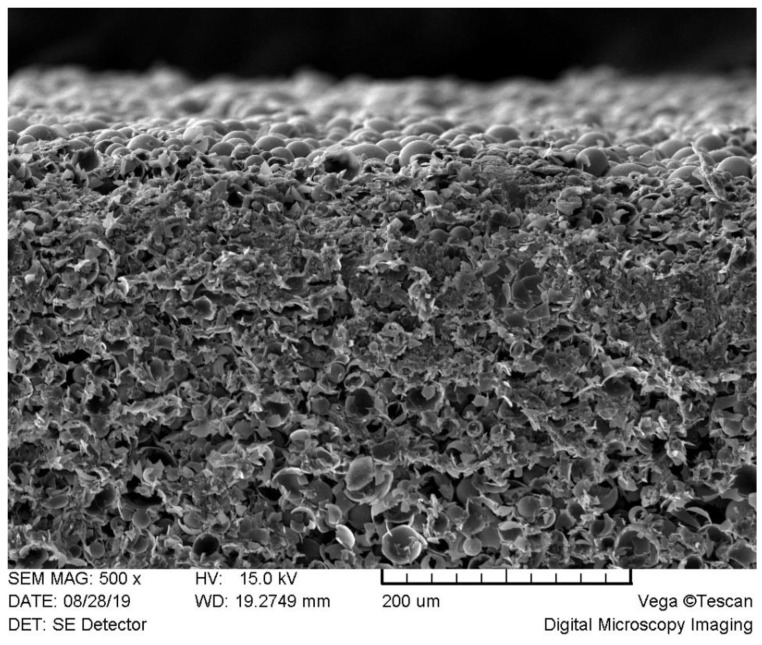
SEM of a cross-section of an exemplary thermo-reflective coating.

**Figure 5 materials-15-05642-f005:**
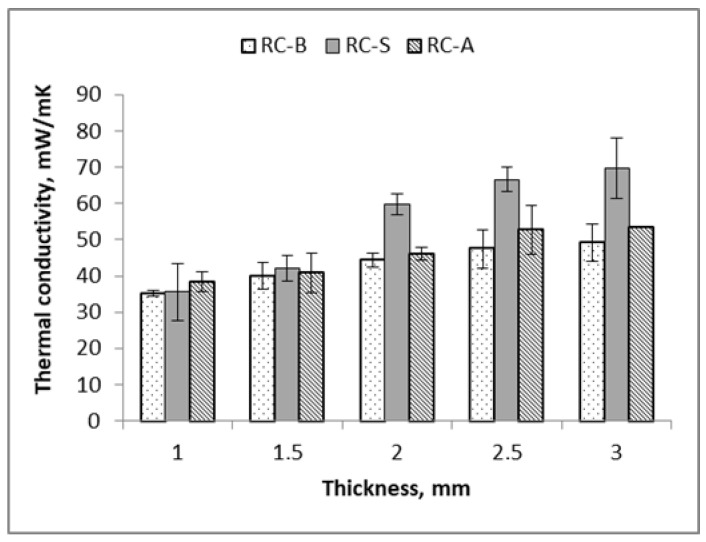
Thermal conductivity of thermal-reflective coatings depending on the thickness of samples.

**Figure 6 materials-15-05642-f006:**
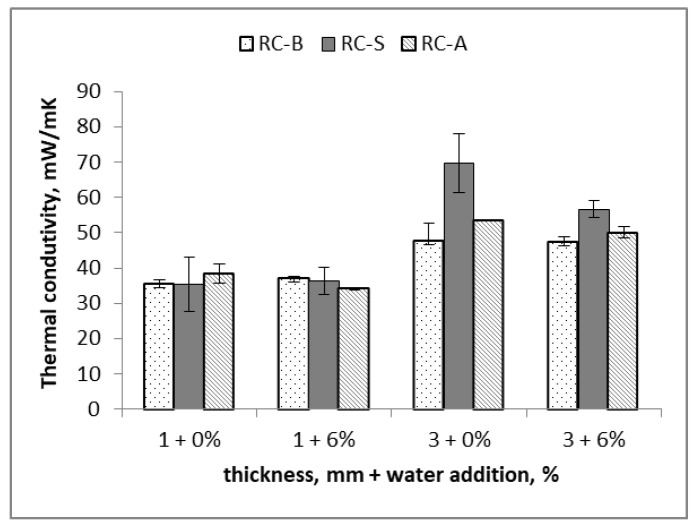
Thermal conductivity depending on the thickness of the RC-B, RC-S or RC-A coating samples for the pastes diluted with water.

**Figure 7 materials-15-05642-f007:**
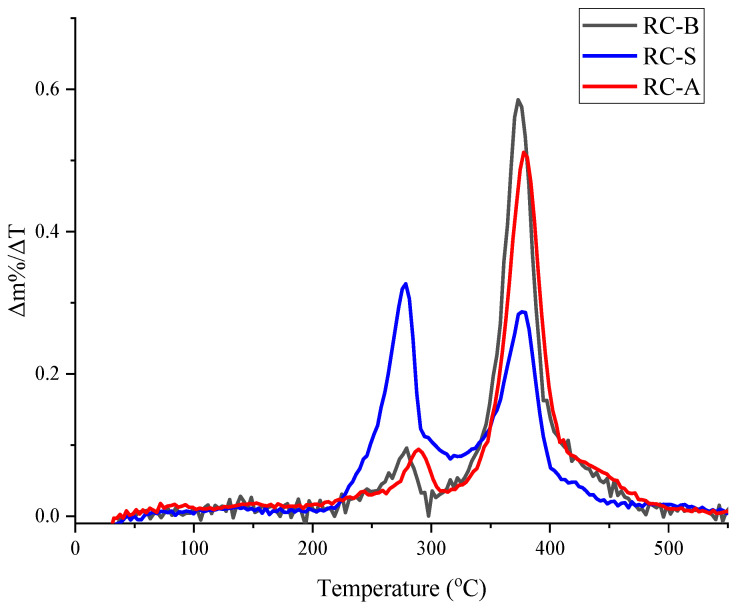
First derivative curve of TGA (tests in a nitrogen media).

**Figure 8 materials-15-05642-f008:**
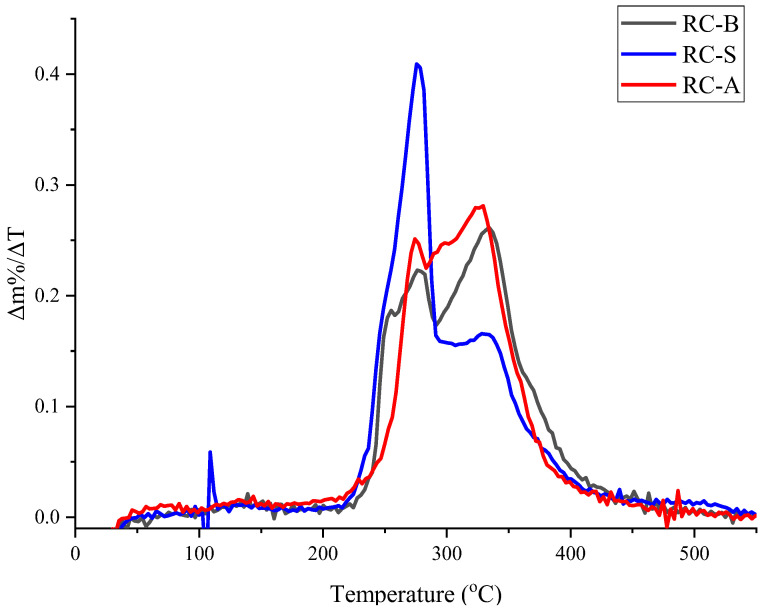
First derivative curve of TGA (tests in the air media).

**Figure 9 materials-15-05642-f009:**
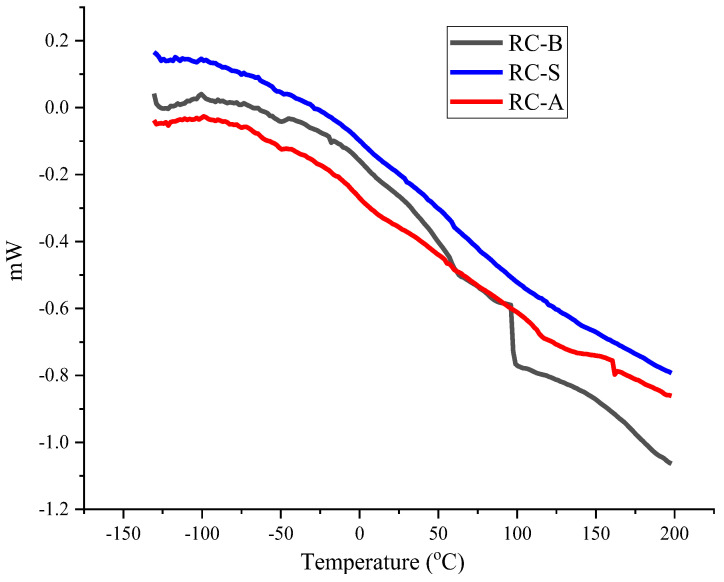
DSC curves of tested thermo-reflecting coatings.

**Figure 10 materials-15-05642-f010:**
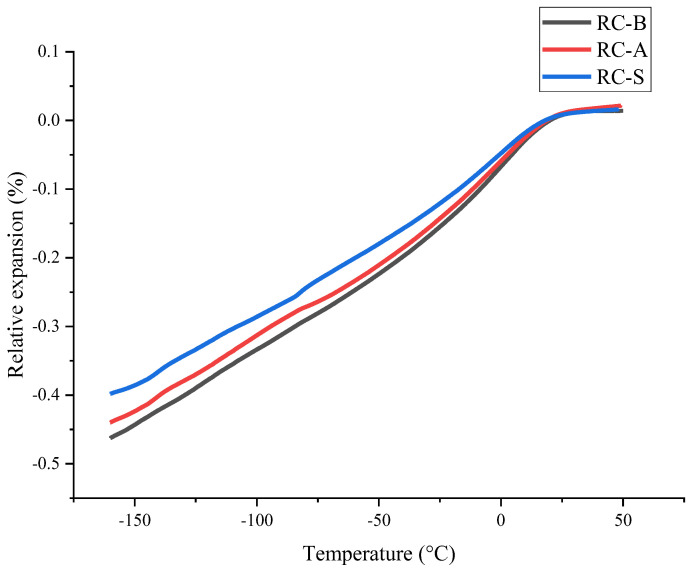
Relative expansion curves of different thermo-reflecting coatings.

**Figure 11 materials-15-05642-f011:**
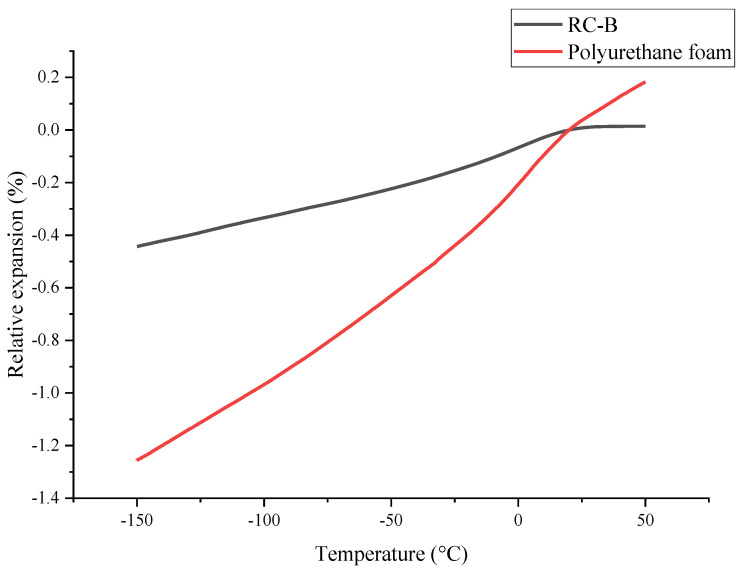
Relative expansion curves of basic thermo-reflecting coating and polyurethane foam.

**Figure 12 materials-15-05642-f012:**
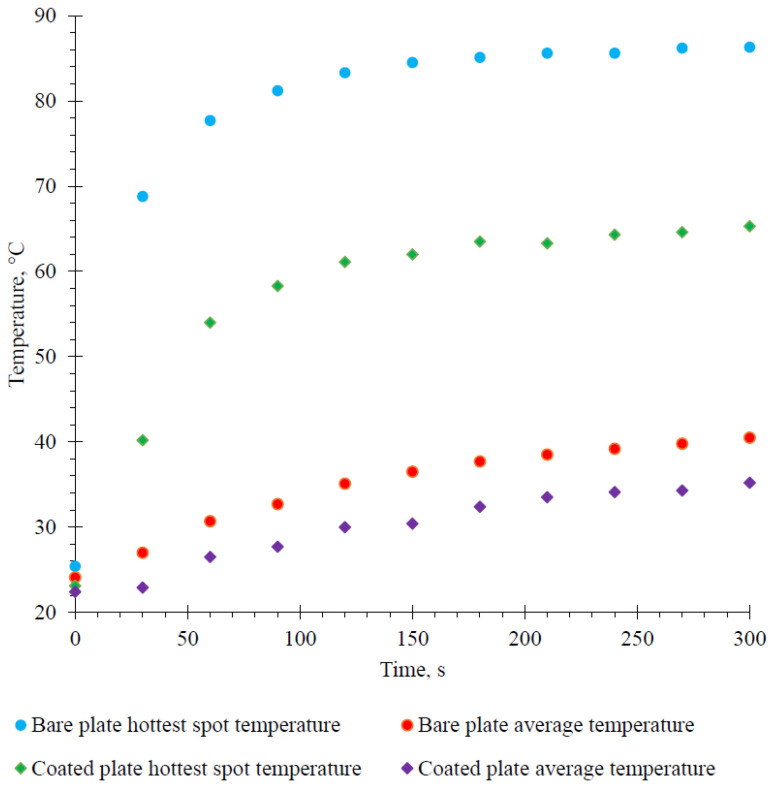
The hottest spot and average temperatures of uncoated and coated aluminum plates vs. heating time.

**Figure 13 materials-15-05642-f013:**
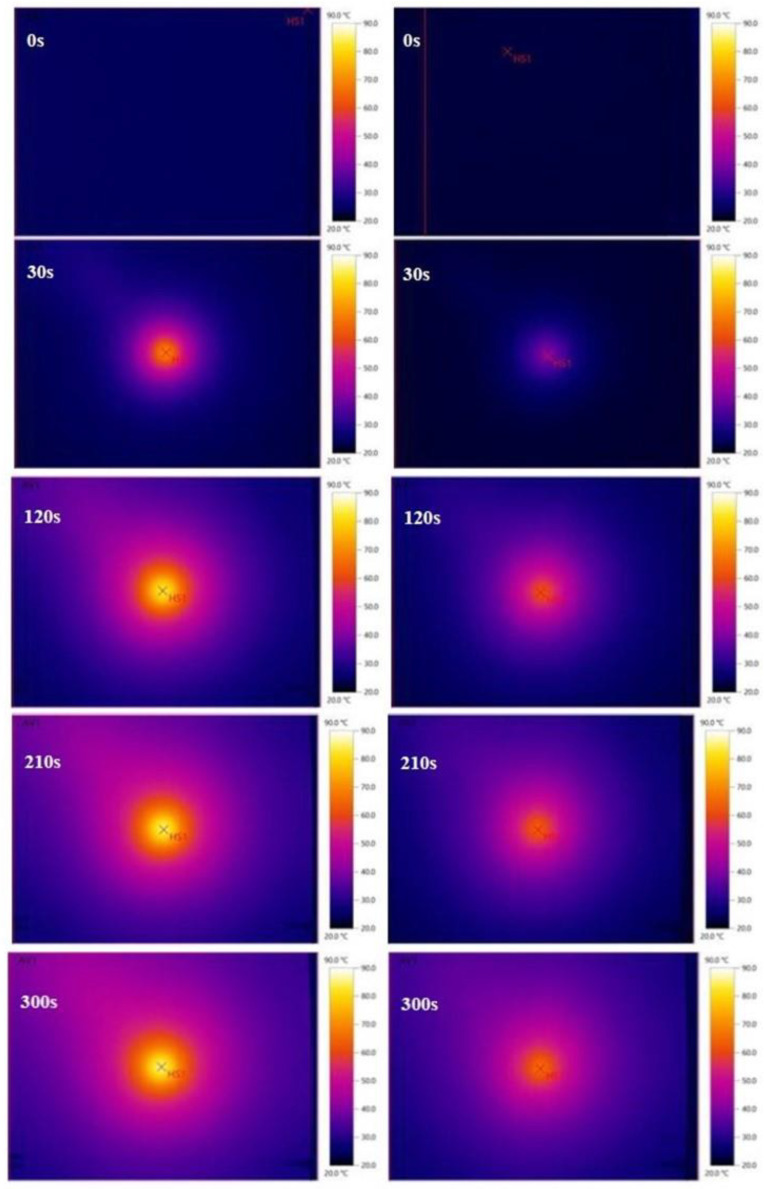
Thermal images of uncoated (**left**) and coated (**right**) aluminum plates during the period 0–300 s of heating.

**Figure 14 materials-15-05642-f014:**
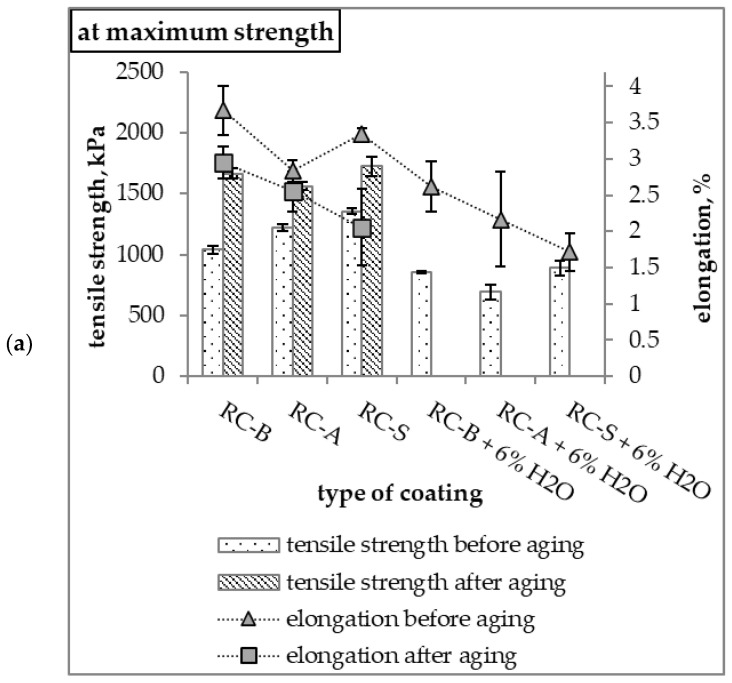
(**a**) Tensile strength and elongation at maximum strength, (**b**) tensile strength and elongation at break.

**Figure 15 materials-15-05642-f015:**
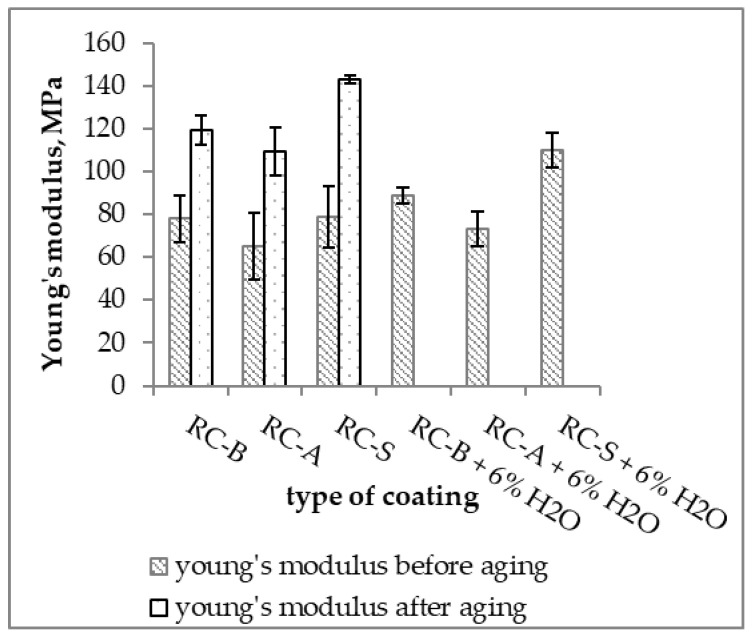
Young’s modulus of coatings.

**Figure 16 materials-15-05642-f016:**
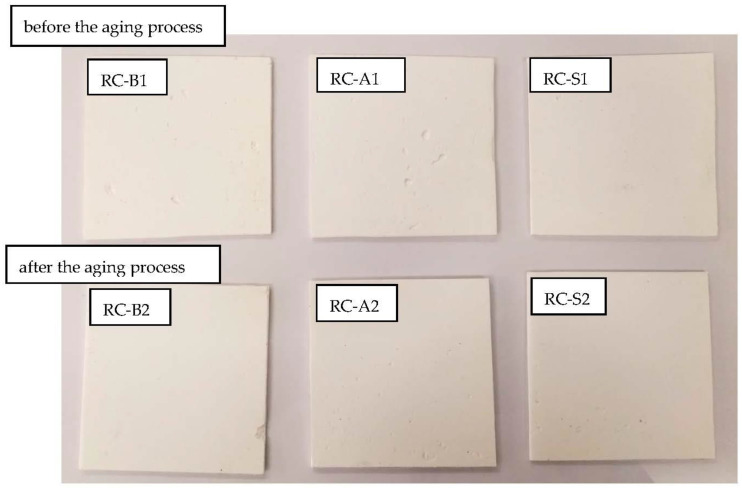
Coating samples before (RC-B1, RC-A1, RC-S1) and after the aging process (RC-B2, RC-A2, RC-S2).

**Figure 17 materials-15-05642-f017:**
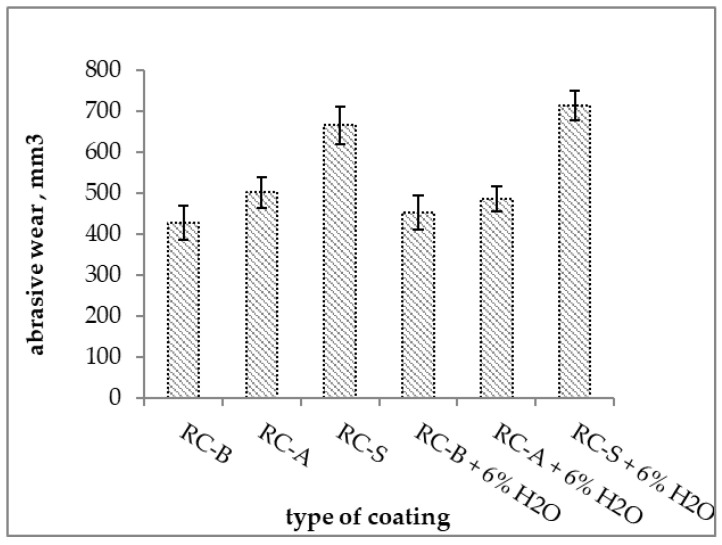
Abrasive wear of tested thermo-reflective coatings.

**Figure 18 materials-15-05642-f018:**
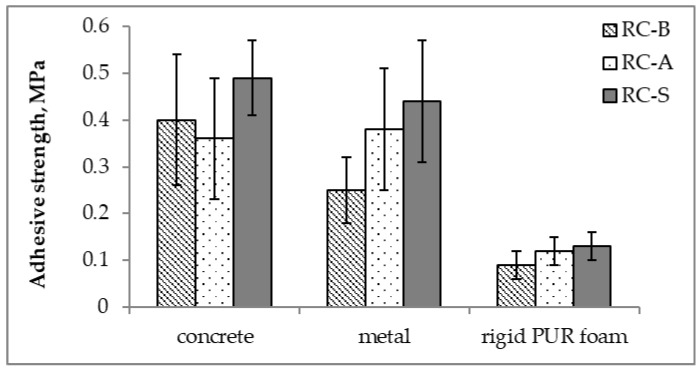
Adhesive strength of coatings.

**Figure 19 materials-15-05642-f019:**
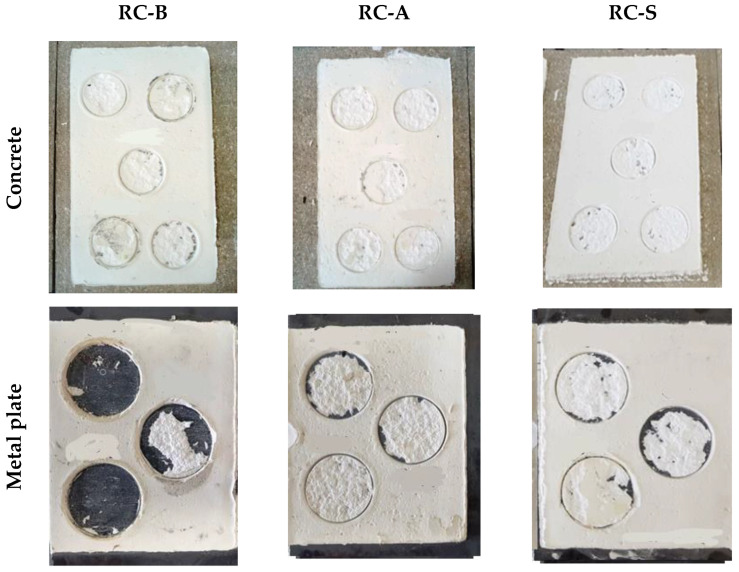
The appearance of the join surfaces after the pull-off test.

**Table 1 materials-15-05642-t001:** Characteristic of tested compositions.

	Paste Type		
	RC-B	RC-A	RC-S
apparent density [g/cm^3^]	0.50–0.55	0.50–0.55	0.65–0.70
pH	9	9	9
dispersion concentration [%]	<25	<25	<25
the content of light fillers [%]	<28	<28	<20
consistency	paste	paste	paste

**Table 2 materials-15-05642-t002:** Thermal stability data obtained by TGA analysis of thermo-reflective coatings.

Coating Sample	T_OS_, °C	T_5_, °C	T_10_, °C	T_15_, °C	T_20_, °C	T_max_, °C	Mass Residue at 750 °C, %
RC-B in N_2_	250	322	363	373	381	280/374	68.6
RC-B in air	225	269	293	319	339	283/335	70.0
RC-S in N_2_	215	270	286	335	370	278/376	69.4
RC-S in air	215	262	277	292	325	279	70.0
RC-A in N_2_	255	301	363	376	386	292/382	68.8
RC-A in air	220	280	301	321	339	286/337	70.3

**Table 3 materials-15-05642-t003:** Glass transition temperatures of different thermo-reflective coatings.

	Thermo-Reflecting Coating Type		
	RC-B	RC-S	RC-A
Tg, °C	97.5	Not detected	161.1

**Table 4 materials-15-05642-t004:** Relative expansion and CTE of different thermo-reflecting coatings.

Thermo-Reflecting Coating Type	Relative Expansion, % at −160 °C	CTE, 10^−6^/°C at −160 °C
RC-B	−0.455 ± 0.008	25.1 ± 0.4
RC-S	−0.383 ± 0.013	21.3 ± 0.7
RC-A	−0.441 ± 0.007	24.5 ± 0.5
Polyurethane foam	−1.446 ± 0.024	80.2 ± 1.5

**Table 5 materials-15-05642-t005:** Apparent density, water absorption, water vapor diffusion resistance factor and dimensional stability of the coatings.

	Apparent Density of Coating [kg/m^3^]	Water Absorption [*v*/*v* %]	OI, %	Water Vapor Diffusion Resistance Factor µ	Dimensional Stability of Coatings, %
RC-B	288.0	21.3	22.7	217	<1
RC-A	277.8	21.9	23.3	126	<1
RC-S	386.9	22.6	40.9	148	<1

## Data Availability

Not applicable.

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
