# Peer review of "New Thermo-Reflective Coatings for Applications as a Layer of Heat Insulating Materials"

_materials, 2022, doi:10.3390/ma15165642_

Round 1
Reviewer 1 Report
This manuscript deals with the characterisation of three types of thermo-reflective coatings based on proprietary formulations, for heat insulation purposes. The topic is interesting and pertinent. The authors put nice efforts in the characterisation of the coatings using different techniques, however, they barely explain the reasons underlying their properties and behaviour differences. Moreover, although one can understand that some specifics of the coating formulation might not be revealed, the authors are almost entirely secretive about their base chemistry, even excluding the silicon dioxide source which is the base component of the coatings. In my opinion, the authors should correlate the coatings properties based on their chemistry, explaining the results obtained in each experiment.
I also have further comments for improvement.
1. In line 39, the authors should explain what they refer to with ‘life cycle’;
2. The portion of the text in lines 92 to 101 describe what was done in this study and, as such, is more compatible with ‘Materials and Methods’ that with ‘Introduction’;
3. In line 119, authors should specify what mean by ‘room conditions’ (i.e. temperature, pressure,…?);
4. I recommend the use of an illustration to aid in describing the thermal transmittance experiment;
5. In line 223, the authors do likely mean ‘mW’ instead of ‘kW’;
6. In table 2, the authors should be clearer about the meaning of ‘Tos’, and ‘T5’ to ‘T20’;
7. The charts in figures 11 and 12 have the word ‘strength’ misspelled in all occurences;
8. I strongly recommend a detailed proofreading of the document and improve text formatting. (e.g. lines 45, 66, 86, 130, 131, 134, 221, 232, 233). Sometimes it is not clear what the authors mean.
Author Response
Dear Rewiver,
Thank you very much for the submitted review and comments on the article. We responded to all comments in the answer below. The changes were introduced in the manuscript and marked in color.
- The authors put nice efforts in the characterisation of the coatings using different techniques, however, they barely explain the reasons underlying their properties and behaviour differences.
Both the scientific and technical literature on the described and characterized coatings is poor. Therefore, it is difficult to relate to the results of other authors. In the revised version, comments on selected results have been supplemented in order to deepen their analysis.
- Moreover, although one can understand that some specifics of the coating formulation might not be revealed, the authors are almost entirely secretive about their base chemistry, even excluding the silicon dioxide source which is the base component of the coatings. In my opinion, the authors should correlate the coatings properties based on their chemistry, explaining the results obtained in each experiment.
Information on the main ingredients of the coatings has been added with the name of the manufacturer.
- In line 39, the authors should explain what they refer to with ‘life cycle’;
Life cycle cost has been clarified
- The portion of the text in lines 92 to 101 describe what was done in this study and, as such, is more compatible with ‘Materials and Methods’ that with ‘Introduction’;
The text has been changed to emphasize the innovation of the conducted research.
- In line 119, authors should specify what mean by ‘room conditions’ (i.e. temperature, pressure,…?);
Room conditions have been clarified in the text.
- I recommend the use of an illustration to aid in describing the thermal transmittance experiment;
The ilustrations describing the thermal transmittance experiment have been added.
- In line 223, the authors do likely mean ‘mW’ instead of ‘kW’;
The error has been corrected
- In table 2, the authors should be clearer about the meaning of ‘Tos’, and ‘T5’ to ‘T20’;
The meaning of the above symbols has been added and explained in the text.
- The charts in figures 11 and 12 have the word ‘strength’ misspelled in all occurences;
The errors have been corrected
- I strongly recommend a detailed proofreading of the document and improve text formatting. (e.g. lines 45, 66, 86, 130, 131, 134, 221, 232, 233). Sometimes it is not clear what the authors mean.
The manuscript have been improved.
Reviewer 2 Report
The paper presents some research gap but novelty is still not clear. At the end of the introduction the novelty should be made clear.
Section 2.3 is too long, please break it down with few subsections based on the charatcerisation techniques. Same for section 3
from the FTIR analysis, please comment on the difference between the three types of coatings
Figure 11a, Strength is a typo mistake
Section 3 should be Results and Discussion
Is there any image or information on cross-sectional morphology of the coatings at different thickness and the thickness uniformity?
In conclusions you should comment on the ranking of the coatings in terms of performance
Figure labels need to be appropriately formatted for example Figure 13
It was evident that different functional performance analysis results are presented but interpretation of the results are lacking. Also comparing and contrasting with the literature is not given.
A mapping of the three coatings based on their performance may be in a table could be presented at the end of Section 3 to better understood the difference between the coatings
English corrections by an experienced colleague or native English speaker must be done.
Author Response
Dear Rewiver,
Thank you very much for the submitted review and comments on the article. We responded to all comments in the answer below. The changes were introduced in the manuscript and marked in color.
- The paper presents some research gap but novelty is still not clear. At the end of the introduction the novelty should be made clear.
The ending of the introduction has been changed to emphasize the innovative nature of the research.
- Section 2.3 is too long, please break it down with few subsections based on the charatcerisation techniques. Same for section 3
Sections 2.3 and 3 have been split into 4 subsections to make the article more readable.
- from the FTIR analysis, please comment on the difference between the three types of coatings
The described peaks are the most intense and come from the basic components found in all the pastes described, therefore there are no significant differences between the analyzed coatings.
- Figure 11a, Strength is a typo mistake
The error has been corrected
- Section 3 should be Results and Discussion
The name of section 3 has been corrected
- Is there any image or information on cross-sectional morphology of the coatings at different thickness and the thickness uniformity?
The SEM image example of cross-sectional morphology as well as a short comment have been added.
- In conclusions you should comment on the ranking of the coatings in terms of performance
The conclusions include the properties of the analyzed coatings. It is impossible to say which one is the best, as the modifications give them other desired properties, i.e. self-extinguishing or anti-corrosive. The choice of a suitable coating will depend on its intended use.
- Figure labels need to be appropriately formatted for example Figure 13
The labels in the picture have been corrected.
- It was evident that different functional performance analysis results are presented but interpretation of the results are lacking. Also comparing and contrasting with the literature is not given.
It is difficult to find references to this type of research in the professional literature, because researchers mainly focus on describing thermo-reflexive properties, temperature and transmittance measurements, ignoring the operational properties of coatings.
- A mapping of the three coatings based on their performance may be in a table could be presented at the end of Section 3 to better understood the difference between the coatings
The conclusions have been refined to make the comparison of the materials clearer. We want to avoid duplicating the same results in the next new table.
Round 2
Reviewer 1 Report
Most of my comments have been addressed. The manusctipt has been improved.
Reviewer 2 Report
all suggested changes are done